# Pericytes as the Orchestrators of Vasculature and Adipogenesis

**DOI:** 10.3390/genes15010126

**Published:** 2024-01-19

**Authors:** Caroline de Carvalho Picoli, Alexander Birbrair, Ziru Li

**Affiliations:** 1Center for Molecular Medicine, MaineHealth Institute for Research, Scarborough, ME 04074, USA; caroline.picoli@mainehealth.org; 2Department of Dermatology, University of Wisconsin-Madison, Medical Sciences Center, Madison, WI 53706, USA; abirbrair@dermatology.wisc.edu

**Keywords:** pericytes, hallmarks, adipose tissue, adipogenesis

## Abstract

Pericytes (PCs) are located surrounding the walls of small blood vessels, particularly capillaries and microvessels. In addition to their functions in maintaining vascular integrity, participating in angiogenesis, and regulating blood flow, PCs also serve as a reservoir for multi-potent stem/progenitor cells in white, brown, beige, and bone marrow adipose tissues. Due to the complex nature of this cell population, the identification and characterization of PCs has been challenging. A comprehensive understanding of the heterogeneity of PCs may enhance their potential as therapeutic targets for metabolic syndromes or bone-related diseases. This mini-review summarizes multiple PC markers commonly employed in lineage-tracing studies, with an emphasis on their contribution to adipogenesis and functions in different adipose depots under diverse metabolic conditions.

## 1. Introduction

Pericytes (PCs) were initially described in the 19th century by Charles-Marie Benjamin Rouget, who observed a population of contractile cells in small blood vessels, and designated them Rouget cells [1]. In the 20th century, Karl Wilhelm Zimmermann renamed these cells “pericytes”, due to their distinct anatomical position around the vasculature [2]. Characterized by long cytoplasmic processes around blood vessel walls [3], PCs communicate with endothelial cells through physical contact or through secreting paracrine signals [4]. Adipose tissues are well vascularized and harbor a significant population of PCs [5] (Figure 1); understanding the functions of these cells and their connections with vascular cells and adipocytes, as well as their effects on adipogenesis, may provide a promising opportunity for therapeutic intervention in metabolic diseases, such as obesity and type 2 diabetes.

Although advances have been made in understanding the effects of PCs on adipose tissue homeostasis, ongoing debates persist on this subject. A primary challenge is that PCs are very heterogeneous, which complicates the characterization of this cell population due to the absence of distinct cell-specific markers. Furthermore, PCs have been implicated in diverse physiological processes, including angiogenesis, tissue regeneration, and adipocyte differentiation, all of which are still under active investigation. Hence, this review will summarize the most recent insights about PC markers based on the lineage tracing studies, and PC functions in adipose tissues, such as white (WAT), brown (BAT), beige and bone marrow (BMAT) adipose tissues.

## 2. Literature Search Strategy

To provide a comprehensive overview of the current state of knowledge regarding the topic of PCs in adipose tissue, relevant papers were chosen from prominent databases, including PubMed/MEDLINE, Google Scholar, Scopus, and Web of Science, covering the past 30 years. An exception was made for the historical review of PCs, extending over 150 years ago. The following search keywords were used: “Pericytes” OR “perivascular cells” AND “adipose tissue” OR “adipocytes” OR “adipogenesis” OR “fat”. Data in each selected paper was carefully reviewed to provide a precise and comprehensive overview of the key aspects of PCs, particularly their contribution and functions in different adipose tissues.

## 3. The Hallmarks of Pericytes: Insights from Lineage-Tracing Studies

It has been challenging to define PCs by a particular molecular marker due to the fact that PC subtypes vary across tissues [6,7,8,9], coupled with differences in the characteristics, functions, and locations of this cell population in a tissue-dependent manner [10]. For example, platelet-derived growth factor receptor-beta (PDGFRβ) [11,12,13,14,15], nerve/glial antigen 2 (NG2, also known as chondroitin sulfate proteoglycan 4, CSPG4) [12,16,17], clusters of differentiation 146 (CD146) [12,18,19,20,21,22], T-box18 (TBX18) [15], α-smooth muscle actin (α-SMA) [10,23], and NESTIN [24,25] are commonly used as PC markers in most of the tissues. However, other tissue-specific markers were also proposed by a single-cell transcriptome analytic study: such as *Kcnk3* in the lung, *Rgs4* in the heart, myosin heavy chain 11 (*Myh11)* and *Knca5* in the kidney, *Pcp4l1* in the bladder, and *Higd1b* in the lung and the heart [26].

In addition to the tissue-specific manner, the identification of PCs varies between species. For example, CD105, CD13 [27], and CD73 [28] were used as PC markers in human studies, while smooth muscle myosin [29], tropomyosin [29], vimentin [30], desmin [10,30], connexin 43 (Cx43) [31], Endosialin (CD248) [32,33], γ-glutamyl transpeptidase (GGT) [34], protein kinase G (PKG) [35], aminopeptidase A (APA) [36], a regulator of G protein signaling 5 (RGS5) [37,38], ATP-sensitive potassium Kir6.1 [39,40], sulfonylurea receptor 2 (SUR2) [41], and delta-like 1 homolog (DLK1) [42,43] were identified in mouse PCs.

However, one of the caveats of these markers is that they are also expressed, to some extent, in other cell types, such as smooth muscle and interstitial cells [44,45,46]. Thus, few dynamic molecular markers could be commonly used in PC identification, with expression varying in a tissue-specific manner or depending on the developmental stages or type of blood vessels. Nevertheless, several PC markers, including PDGFRβ, NG2, TBX18, SMA, NESTIN, and CD146, have been used for PC profiling and lineage-tracing studies to identify the involvement of PCs in adipocyte differentiation. We will summarize the contribution of these PCs labelled by different markers in WAT and BAT under different metabolic or environmental conditions (Table 1).

PDGFRβ—Using a pulse-chase lineage-tracing approach, Shao et al. found that PDGFRβ^+^ preadipocytes contribute to gonadal and perirenal WAT expansion upon high-fat diet feeding, but not in inguinal WAT [51]. In addition, *Pparg* overexpression in PDGFRβ^+^ mural cells give rise to healthier white adipocytes under high-fat diet-induced obesity. Furthermore, Zfp423^+^ PDGFRβ^+^ cells, which are the PCs that are committed to adipogenic lineage, contribute to approximately 10% of the white adipocytes in gonadal WAT following high-fat diet feeding. Moreover, these cells demonstrate the ability to differentiate into beige adipocytes in small clusters under prolonged cold exposure for two weeks; of note, this occurs in multiple waves depending on the extent and time course of cold stress [14]. A recent study focusing on BAT demonstrated that PDGFRβ^+^ PCs gave rise to brown adipocytes at an early stage, but not in adulthood [15].

NG2—It is found to be expressed on the surface of PCs during vasculogenic and angiogenic processes [52,53], as well as on some adipocytes. A lineage-tracing study demonstrated that NG2^+^ cells marked a portion of the mural cells and 100% of white adipocytes in subcutaneous and periscapular regions under room temperature, but only tracked mural cells in the visceral depot vasculatures without signals in adipocytes [23]. Interestingly, under cold exposure, approximately 50% of the UCP1^+^ beige cells originated from NG2^+^ PCs in subcutaneous and visceral perigonadal depots, suggesting the existence of other potential progenitor sources contributing to beige adipocytes [23]. However, NG2^+^ cells do not contribute to intramyocardial adipocytes in postnatal development or under adult homeostasis [47].

TBX18—It serves as a robust marker for identifying mural cells, including PCs and vascular smooth muscle cells [48]. Lineage-tracing mouse studies demonstrated that TBX18 neither traced to white adipocyte under either chow or high-fat diets, nor did it lead to a significant contribution to beige cells after 14 days of treatment with a β3-adrenergic receptor agonist [48]. Likewise, another study demonstrated that TBX18 plays a minor role in de novo beige adipogenesis [54]. However, in BAT, TBX18^+^ PCs act as progenitors of brown adipocytes in multiple regions, such as thoracic perivascular and supraclavicular depots, but not in intrascapular and periaortic regions, suggesting that TBX18^+^ PCs contribute to brown adipocytes in a depot-specific manner [15].

α-SMA—Similar to TBX18, α-SMA was not expressed in mature adipocytes, but was restricted to vasculatures in subcutaneous and visceral adipose tissue under normal conditions. However, upon cold exposure for 7 days, α-SMA^+^ PCs were fated into de novo beige adipogenesis and contributed to 55–68% of beige adipocytes in inguinal and perigonadal adipose depots [23].

NESTIN—It labels a subset of pericyte-like cells that possess somatic stem/progenitor cell properties [55]. Iwayama T. et al. [24] demonstrated that NESTIN was expressed in the PCs in WAT; and these NESTIN^+^ PCs have the potential to differentiate into either adipocytes or profibrotic cells. Especially, when PDGFRα^+^ was overexpressed in NESTIN^+^ PCs, these cells were fated into profibrotic cells by increasing the production of collagen [24], while the adipogenesis was inhibited.

CD146^+^—CD146^+^CD31^−^CD45^−^ PCs from adipose tissue prefer to differentiate into adipocytes, while the periosteal CD146^+^ PCs tend to be more mineralized in vitro and ossification in vivo [49]. These results suggest that CD146^+^ PCs hold a tissue-specific potential of differentiating into either adipogenic or osteogenic cells. A key regulator in this fate determination of CD146^+^ PCs is the T cell lymphoma invasion and metastasis 1 (TIAM1), which is highly expressed in adipose tissue but not in the skeleton. TIAM1 affects the morphology of adipose tissue-derived CD146^+^ PCs and increases the potential for adipogenic differentiation, while TIAM1 knockdown promotes osteogenic differentiation over adipogenesis [50].

As discussed earlier, none of the single molecular markers could uniquely identify PCs in general, or effectively distinguish them from other cell types, such as mural cells and fibroblasts. Future studies utilizing the single-cell RNA-seq technique will provide more informative data, shedding light on the heterogeneity of this cell population [26,56,57]. Furthermore, it remains unclear whether these PC progenitors would influence adipocyte dynamics under conditions such as physical exercise, diabetes, hypertension, fatty liver disease, and other metabolic complications [58,59].

## 4. Pericytes in the Bone Marrow Niche

Bone is also highly vascularized, with the blood vessels presenting throughout the bone tissue except in cartilaginous areas such as the growth plate [60,61,62,63]. The vasculature is a key component of the bone marrow microenvironment. Of note, PCs around vasculatures provide critical signals for bone marrow niche homeostasis, such as stromal cell proliferation, hematopoietic stem cell (HSC) maintenance, as well as the regulation of osteoblast, osteoclast, and adipocyte differentiation [64,65].

Two subtypes of vessels have been classified in the bone marrow based on their marker expression and functional characteristics, including type-H, which is localized in the proximal metaphyseal regions with higher blood flow, and type-L, which is mainly located in the diaphyseal regions and consists predominantly of sinusoidal-like vessels [66,67]. The distribution of PCs varies between these two distinct types of bone marrow vessels. For example, type-H capillaries are surrounded by perivascular cells expressing PDGFRβ and NG2 [68], whereas type-L blood vessels are surrounded by perivascular cells expressing leptin receptor (LepR) [69] and reticular cells abundant in CXCL12 (CAR) [70].

PDGF-PDGFRβ signaling in Osterix-expressing cells are critical for the recruitment, expansion, and angiotropism of skeletal stem and progenitor cells during bone repair, which are essential processes for effective fracture healing [71]. NG2^+^ PCs are critical for HSC localization to arterioles and the maintenance of their quiescent status [72]. LepR^+^ cells have the potential to differentiate into osteoblasts, chondrocytes, and adipocytes [73]. Notably, LepR labels 70% and 90% of bone marrow adipocytes at 2 and 6 months of age, respectively [73]. Although LepR^+^ cells barely contribute to bone formation when mice were younger than 2 months old, this proportion increases with age, reaching up to ~80% when mice were 14 months old [73]. Of note, leptin-lepR signaling in skeletal stem cells inhibits osteogenesis and promotes adipogenesis in response to a high-fat diet [74], and impairs bone regeneration after injury. CAR cells are scattered throughout the bone marrow, creating a network with the blood vessels [70,75]. These cells demonstrated the potential to differentiate into adipocytes both in vitro [76] and in vivo [77]. The pre-adipocyte-like CAR cells are also readily lipid-laden because CXCL12 deletion directly converted these cells into mature bone marrow adipocytes. Moreover, pre-adipocyte-like CAR cells communicate with hematopoietic cells through the CXCL12–CXCR4 pathway to regulate HSC retention, quiescence, and repopulation [77].

LepR^+^ cells and CAR cells overlap with each other to some extent [73,78], and both cell populations express high level of adipogenic markers, such as *Pparg*, *Adipoq*, and *Cebpa* [73,76]. Therefore, Zhong et al. described that LepR^+^/CAR cells could be marrow adipogenic lineage precursors (MALPs) [79], which were identified in the bone marrow niche through a single-cell RNA-sequencing approach [80]. MALPs express adipogenic markers (such as *Pparg*, *Cebpa*, *Adipoq*, *Apoe*, and *Lpl*), without lipid accumulation (*Perilipin*^−^) [79]. Interestingly, MALPs also express PCs markers, including PDGFRβ and Laminin [80], and are exhibited as star-shaped PCs, with many dendritic processes extending and connecting around the endothelial wall [80].

Although bone marrow PCs express PDGFRβ, NG2, LepR, and/or CXCL12, these markers are not exclusive to PCs, as they are also expressed in stromal cells and/or MALPs. This complexity leads to questions about how to distinguish the typical PCs from bone marrow stromal cells and adipogenic MALPs, which require further studies. Overall, these findings suggest that PCs have a significant impact on adipogenesis in the bone marrow and influence stromal cells proliferation, HSC maintenance, and other mesenchymal cells.

## 5. Pericyte Functions in Adipose Tissue

As described above, the existence of PCs in adipose tissue has drawn more attention from researchers, prompting the functional studies of this cell population both locally and systematically. Overall, the primary functions of PCs in adipose tissue include maintaining vasculature integrity, facilitating angiogenesis, controlling blood flow, and serving as a stem/progenitor cell pool (Table 2).

Vasculature integrity—PCs intimately envelop endothelial cells in capillaries and microvessels [85], and are therefore predominantly found in the abluminal wall of blood vessels, where they contribute to the maintenance of capillary integrity and vascular permeability [81]. The basal membrane of the PC forms a continuous connection with the endothelial cells; both cell populations could secrete extracellular matrix proteins (mainly collagen IV and glycoprotein laminin) to maintain the structural integrity of blood vessels. The PCs could also emit protrusions that insert into the endothelial cell invaginations (cavities) and occasional interruptions of the basement membrane, providing essential support and structure for cell–cell communications [82,83]. Beyond their roles in regulating or stabilizing the function of blood vessels, PCs could be disrupted by pathological processes, and participate in vascular tissue remodeling [81,82,83,84].

It is noteworthy that the density of the endothelial cell/PC ratio varies across tissues, suggesting that PCs may perform specialized functions that differ between organs. The highest ratio of endothelial cells versus PCs is found in the central nervous system and retina [82,106,107], approximately 1:1, compared with other organs and tissues such as kidney (2.5:1), lung (7–9:1), skin, liver (10:1), cardiac microvasculature (2–3:1), and skeletal muscle (100:1) [108,109,110,111]. However, the proportion of PCs in adipose tissue remains undetermined.

Angiogenesis—PCs are the first cells to invade newly vascularized tissues, determining the location of sprout formation [86,87,88]. They actively participate in recruitment, extracellular matrix modulation, and paracrine signaling, and direct interactions with endothelial cells [112]. PC proliferation and migration is closely coordinated with the behavior of adjacent endothelial cells, demonstrating the essential collaboration between both cell types during angiogenesis. Indeed, the co-culture of human adipose tissue-derived CD146^+^ PCs with human umbilical vein endothelial cells (HUVECs) demonstrated a clear promoting effect on vessel sprouting during the angiogenesis [89]. Furthermore, the pro-angiogenic efficacy of adipose tissue-derived PCs on tube formation and cell migration was enhanced by Nel-like protein-1 (NELL-1), and promoted bone formation in an osteonecrosis mouse model [113]. Moreover, PCs have been found to ameliorate ischemia after being transplanted into a mouse model with severe hind limb ischemia [114,115], and improve the blood flow in mice during femoral artery ligation [116], suggesting their potential as promising targets for vascular regeneration. It is important to note, however, that their heterogeneity implies that different PC subpopulations may have distinct functions based on the tissue of origin and surface markers [117].

Capillary blood flow—The PCs have the abilities to induce vasoconstriction or vasodilation within the capillary beds in order to control the vascular diameter as well as the blood flow, akin to the smooth muscle cells of larger vessels. PCs also express cholinergic and adrenergic receptors (α-2 and β-2), where a β-adrenergic response leads to relaxation, while an α-2 response would be antagonistic and induce contraction. There are also other vasoactive substances that bind to PCs, such as angiotensin II and endothelin 1, and function as paracrine signals to regulate contraction and relaxation [90]. Moreover, using optogenetic approaches, the stimulation of the PCs causes excitation that leads to the contractions of blood vessels in brain [91], whereas Halorhodopsin channel hyperpolarization in PCs, which inhibits cell activities, results in increased capillary blood flow in the retina [92]. The modulation of PC activities provides opportunities to optimize the microvascular network within adipose tissue, thereby improving perfusion and nutrient exchange. This manipulation not only holds potential for mitigating hypoxia within adipose tissue, but also for fostering a more conducive environment for adipocyte function. Optimizing capillary blood flow through PCs may have implications for metabolic health, as it could affect adipose tissue expansion, adipokine secretion, and overall tissue homeostasis.

Stemness property—PCs possess the capacity for self-renewal and differentiation into other cell lineages, such as mature adipocytes, osteoblasts, and other mesenchymal cell types [25,93,94,95,96,97]. Therefore, they have been implicated as a potential reservoir of multipotent stem cells in adults [55,81,93,118,119]. In adipose tissue, PCs exist in the stromal vascular fraction (SVF) [11], which contains cells that are crucial for adipose tissue plasticity, angiogenesis, and neovascularization [120,121]. Notably, it has been demonstrated that PCs in close proximity to vasculature serve as an important source of adipogenic progenitor cells [98,99,100,101,122], which has been discussed in the “hallmarks” section above. In addition, Olson and Soriano found that PDGFRβ activation driven by Sox2-Cre in epiblasts, which include PCs and mesenchymal cells, inhibits white adipocyte differentiation [123]. In contrast, the deletion of PDGFRβ enhances white, brown, and beige adipogenesis [124]. Interestingly, the stemness property of PCs has been applied in reconstructive and tissue engineering therapies. For example, human adipose-tissue-derived stem cells (ASCs) isolated from lipoaspirate, expressing CD146, including PCs and adventitial components, have been demonstrated to have angiogenic and adipogeneic properties in vitro [102]. These cells have also been shown to have the capability to undergo osteogenic differentiation in vitro and form bone in vivo, suggesting their potential as a source for bone formation [103] and the possibility of contributing to bone healing [104]. In addition, a study using human ASC-derived PCs, injected into NOD-SCID mice, demonstrated the enhancement of retinal microvascular stabilization in mouse models of retinopathic vasculopathy [125].

Beige adipocytes have been gaining more attention from biologists, as these cells arise in WAT following thermogenic induction [126]. Interestingly, these beige adipocytes are located around the vasculature, sharing the same location with PCs (Figure 2). A groundbreaking study by Clack and Clack (1940) had revealed that de novo adipogenesis could occur in the close proximity of blood vessels, suggesting that adipogenic progenitors might be a type of blood vessel wall cells [48]. Indeed, perivascular cells (vascular smooth muscle cells and PCs) have been proven to give rise not only to white adipocytes, but also to differentiate into beige adipocytes [47,105,122]. Berry et al. [23] also evaluated a variety of mural cell markers, including SM22, Myh11, NG2, PDGFRα, and SMA, affirming that these cells indeed serve as an important source for beige adipocyte induction during cold exposure. However, further data analysis or future studies are required to better understand which of these markers truly represent PCs in WAT. Despite efforts to elucidate the ideal identity of the adipogenic progenitors among PCs, there are frequent contradictions in the findings, attributed not only to a variety of lineage tracing markers, but also to different experimental conditions [48].

## 6. Conclusions and Prospects

PCs have emerged as promising and multifaceted cells in various adipose tissues, orchestrating a series of functions crucial for adipose tissue homeostasis. Their regulatory roles in adipogenesis and angiogenesis highlight their importance in the formation of the adipose tissue microenvironment. The manipulation of PCs holds great potential to influence capillary integrity, angiogenesis, blood flow, and adipogenesis, thus impacting the metabolic health of adipose tissue. As we delve deeper into understanding the complexities of PC biology, we open new avenues for innovative therapeutic interventions in conditions related to adipose tissue dysfunction, offering hope for addressing issues such as obesity and metabolic disorders. Further research is required to fully understand the mechanisms, triggers, and functional consequences of PC-to-adipocyte differentiation. Continued investigations will shed more light on the role of PCs in adipose tissue biology and the progression of metabolic diseases related to adipose tissue.

## Figures and Tables

**Figure 1 genes-15-00126-f001:**
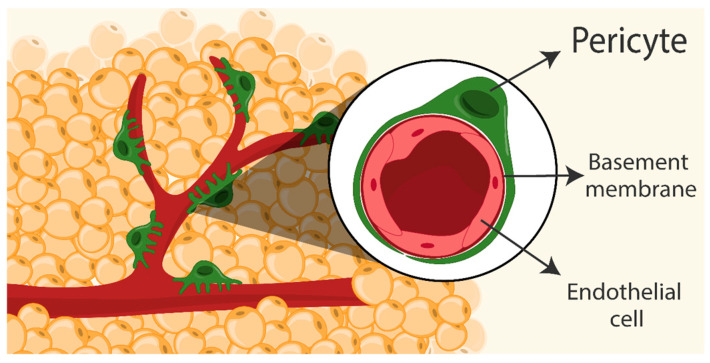
A schematic illustration depicts the location of pericytes within the adipose tissue niche. Adipose tissue is well vascularized, with pericytes surrounding blood vessels. Notably, pericytes are embedded within the shared basement membrane alongside endothelial cells.

**Figure 2 genes-15-00126-f002:**
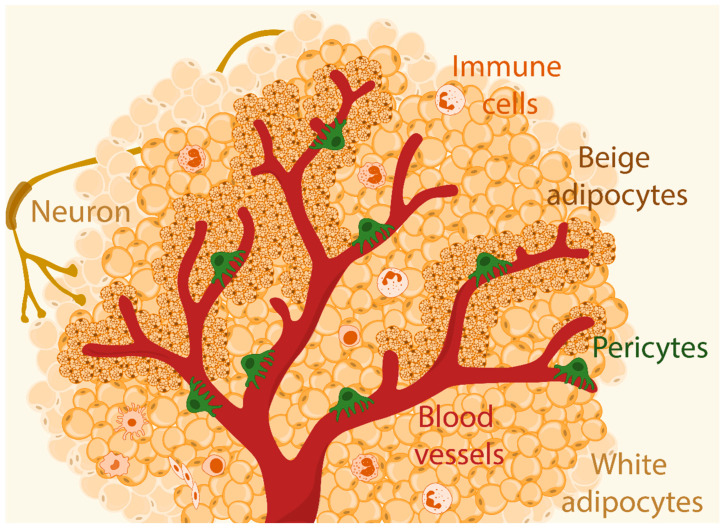
Representative schematic of the white adipose tissue microenvironment, including beige adipocytes surrounding blood vessels, pericytes, immune cells, and neurons in the stromal vascular fraction.

**Table 1 genes-15-00126-t001:** Overview of molecular markers for pericytes in adipose tissues.

Marker	Location	Function in Adipose Tissue	Condition	Specie	References
PDGFRβ	Gonadal WAT	Contribute to gonadal WAT expansion	High-fat die	Mouse	Vishvanath et al. [14]
	Inguinal WAT	Contribute to beige adipocytes formation	Cold exposure	Mouse	Vishvanath et al. [14]
	All BAT depots	Label brown adipocytes in early postnatal stage, but not in adulthood	Different developmental stages	Mouse	Shi et al. [15]
NG2 (CSPG4)	Subcutaneous and periscapular WAT	Trace to 100% white adipocytes	Room temperature	Mouse	Berry et al. [23] Jiang et al. [47]
	Subcutaneous and perigonadal WAT	Track 50% of the UCP1^+^ beige cells	Cold exposure	Mouse	Berry et al. [23]
TBX18	Perigonadal and inguinal WAT	No contribution to white adipocytes	High-fat diet or normal chow diet; β3-adrenergic receptor agonist treatment	Mouse	Cattaneo et al. [48]
	Supraclavicular, thoracic perivascular and perirenal BAT	Act as progenitors of brown adipocytes in a depot-specific manner	Different developmental stages	Mouse	Shi et al. [15]
α-SMA	Subcutaneous and visceral WAT	No contribution to white adipocytes	Room temperature	Mouse	Berry et al. [23]
	Inguinal and perigonadal WAT	Trace to 55–68% of beige adipocytes	Cold exposure	Mouse	
Nestin	Dermal WAT	Potentially become adipocytes or profibrotic cells	Ex vivo	Mouse cell	Iwayama et al. [24]
CD146	Human liposuction specimens	Prefer to differentiate into adipocytes over osteogenic cells	Ex vivo	Human cell	Xu et al. [49] Hsu et al. [50]

**Table 2 genes-15-00126-t002:** Brief summary of pericyte functions.

Function	Brief Description	References
Vasculature integrity	Envelop endothelial cells; Secrete extracellular matrix proteins; Provide structural support to blood vessels.	Nwadozi et al. [81], Armulik et al. [82], Holm et al. [83], Birbrair et al. [84], and Andreeva et al. [85]
Angiogenesis	Determine the location of sprout formation; Support the angiogenic sprouting process; Coordinate with the adjacent endothelial cells and promote angiogenesis.	Nehls et al. [86], Ponce et al. [87] Eilken et al. [88] and J. Gonzalez-Rubio et al. [89]
Capillary blood flow	Regulate vascular diameter and blood flow; Modulate blood vessel contraction and relaxation.	Rucker et al. [90], Nelson et al. [91] and Ivanova et al. [92]
Stemness property	Possess self-renewal and differentiation capacity; Contribute to adipose tissue plasticity, angiogenesis, neovascularization, and osteogenesis; Give rise to white, brown and beige adipocytes.	Mendez-Ferrer [25], Crisan et al. [93], Hoshino et al. [94], Passman et al. [95], Corselli et al. [96], Pierantozzi et al. [97], Rodeheffer et al. [98], Farrington-Rock et al. [99], Lin et al. [100], Cai et al. [101], Lauvrud et al. [102], James et al. [103] Wang et al. [104], Berry et al. [23]; Jiang et al. [47], Cattaneo et al. [48] and Shao et al. [105]

## Data Availability

Not applicable.

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
