# Peer review of "Pericytes as the Orchestrators of Vasculature and Adipogenesis"

_genes, 2024, doi:10.3390/genes15010126_

Round 1

Reviewer 1 Report

Comments and Suggestions for Authors

The manuscript reviewed the pericyte as the orchestrator of vasculature and adipogenesis. The authors summarise multiple pericyte markers employed in lineage tracing studies, and their contribution to adipogenesis in different adipose depots. They also summarize the pericyte functions in adipose tissue, particularly maintaining vasculature integrity, controlling blood flow, and serving as a stem cell pool. Overall, the manuscript is well written and organized, and presents the important roles of pericytes in adipose tissues. Given the high prevalence of metabolic diseases, such as obesity and type 2 diabetes, and the adipose tissue as the root cause of metabolic diseases, this review is timely important. Thus, I would like to recommend the manuscript for publication.

Specific comments -

It will be better if pericyte markers are summarized in a table.

Pericytes also exert an important role in vascular development and angiogenesis.  It will be better if the authors summarize the role of pericytes in vessel formation, angiogenesis, endothelial cell proliferation, migration, etc.

The authors should define all abbreviations used in the manuscript, for example, NG2.

Author Response

Re: Genes-2807988: “Pericyte as the orchestrator of vasculature and adipogenesis”.

We appreciate reviewers’ creative and insightful thoughts. Please find our point-by-point response in red fonts below:

Reviewer 1:

Comments and Suggestions for Authors

The manuscript reviewed the pericyte as the orchestrator of vasculature and adipogenesis. The authors summarise multiple pericyte markers employed in lineage tracing studies, and their contribution to adipogenesis in different adipose depots. They also summarize the pericyte functions in adipose tissue, particularly maintaining vasculature integrity, controlling blood flow, and serving as a stem cell pool. Overall, the manuscript is well written and organized, and presents the important roles of pericytes in adipose tissues. Given the high prevalence of metabolic diseases, such as obesity and type 2 diabetes, and the adipose tissue as the root cause of metabolic diseases, this review is timely important. Thus, I would like to recommend the manuscript for publication.

Specific comments -

It will be better if pericyte markers are summarized in a table.

Reply: We are grateful of this creative suggestion. We have summarized the pericyte markers in Table 1(Page 3). In addition, we created a Table 2 (Page 5)  to summarize some of the functions of pericytes in adipose tissues.

Pericytes also exert an important role in vascular development and angiogenesis.  It will be better if the authors summarize the role of pericytes in vessel formation, angiogenesis, endothelial cell proliferation, migration, etc.

Reply: We appreciate the reviewer's comment. We added a paragraph to discuss the importance of pericytes in angiogenesis, lines 222 - 237.

The authors should define all abbreviations used in the manuscript, for example, NG2.

Reply: We have gone through the whole manuscript and spelled out all the abbreviation, except some single-used gene names.

Reviewer 2 Report

Comments and Suggestions for Authors

The paper "Pericyte as the orchestrator of vasculature and adipogenesis is a short review about the role of pericytes in the human body."

Some slight changes may improve the paper.

Abstract

First the abstract should be rewritten in order to be more informative. It should be emphasized that this is a state-of-the-art paper (not a systematic review).

Methodology

There is no methodology how the papers cited in the manuscript were selected, an from which period were included. Maybe the authors do not need to provide like in meta analysis the number of papers that have been overviewed but which base such as PubMed or Google Scholar were visited and which key words were applied during the search of the references given in the paper is the minimum data that should be given.

Results and disscussion

After the line Regardless, some commonly used PC markers, including PDGFR , NG2, TBX18, SMA, NESTIN, CD146,  were used for profiling PCs and related lineage-tracing studies to identify the involvement of PCs in adipocyte differentiation. – there should be introduction to what in the following lines is presented. Also I strongly recommend for the authors to give beside the narrative part a Table in which the data presented afterwards are summarized – the name of the marker, their abundance in the tissue, the role and reference. It would be much more apprehensive and suitable for further citation. The same type of table, similar to the previous one should be introduced for the third chapter where the role of the pericytes is given in a narrative way.  I strongly believe that these type of tables given within the paper would present an added value of the paper.  

Author Response

Re: Genes-2807988: “Pericyte as the orchestrator of vasculature and adipogenesis”.

We appreciate reviewers’ creative and insightful thoughts. Please find our point-by-point response in red fonts below:

Reviewer 2 :

Comments and Suggestions for Authors

The paper "Pericyte as the orchestrator of vasculature and adipogenesis is a short review about the role of pericytes in the human body."

Some slight changes may improve the paper.

Abstract

First the abstract should be rewritten in order to be more informative. It should be emphasized that this is a state-of-the-art paper (not a systematic review).

Reply:  We thank Reviewer for this comment to improve our paper. We rewrote the abstract to make it more informative and to emphasize that it was a mini-review, not a systematic review.

Methodology

There is no methodology how the papers cited in the manuscript were selected, an from which period were included. Maybe the authors do not need to provide like in meta analysis the number of papers that have been overviewed but which base such as PubMed or Google Scholar were visited and which key words were applied during the search of the references given in the paper is the minimum data that should be given.

Reply: We would like to the reviewer’s insightful comments. We have added the following paragraph to clarify how we searched the references:

“2. Literature search strategy

To provide a comprehensive overview of the current state of knowledge regarding the top-ic of PCs in adipose tissue, relevant papers were chosen from prominent databases, in-cluding PubMed/MEDLINE, Google Scholar, Scopus, and Web of Science, covering the past 30 years. An exception was made for the historical review of PCs, extending over 150 years ago. The following search keywords were used: “Pericytes” OR “perivascular cells” AND “adipose tissue” OR “adipocytes” OR “adipogenesis” OR “fat”. Data in each se-lected paper was carefully reviewed to provide a precise and comprehensive overview of the key aspects of PCs, particularly their contribution and functions in different adipose tissues.”

Results and discussion

After the line Regardless, some commonly used PC markers, including PDGFR , NG2, TBX18, SMA, NESTIN, CD146,  were used for profiling PCs and related lineage-tracing studies to identify the involvement of PCs in adipocyte differentiation. – there should be introduction to what in the following lines is presented.

Reply: We have added the following sentence to introduce what will be discussed below: “We will summarize the contribution of these PCs labelled by different markers in WAT and BAT under different metabolic or environmental conditions (Table 1).” (lines 82 – 84).

Also I strongly recommend for the authors to give beside the narrative part a Table in which the data presented afterwards are summarized – the name of the marker, their abundance in the tissue, the role and reference. It would be much more apprehensive and suitable for further citation. The same type of table, similar to the previous one should be introduced for the third chapter where the role of the pericytes is given in a narrative way.  I strongly believe that these type of tables given within the paper would present an added value of the paper. 

Reply: We really appreciate these suggestions. We have added a table 1- referring to pericyte markers in adipose tissue and table 2- referring to the functions of pericytes.

Round 2

Reviewer 2 Report

Comments and Suggestions for Authors

The authors have made substantial changes. Thus, the paper may be accepted for publication.